# The Impressive Anti-Inflammatory Activity of Cerium Oxide Nanoparticles: More than Redox?

**DOI:** 10.3390/nano13202803

**Published:** 2023-10-21

**Authors:** Francesca Corsi, Greta Deidda Tarquini, Marta Urbani, Ignacio Bejarano, Enrico Traversa, Lina Ghibelli

**Affiliations:** 1Department of Chemical Science and Technologies, University of Rome “Tor Vergata”, 00133 Rome, Italy; greta.deidda.tarquini@uniroma2.it (G.D.T.); marta.urbani96@gmail.com (M.U.); traversa@uniroma2.it (E.T.); 2Department of Biology, University of Rome “Tor Vergata”, 00133 Rome, Italy; 3Institute of Biomedicine of Seville (IBiS), University of Seville, HUVR, Junta de Andalucía, CSIC, 41013 Seville, Spain; ibejarano@us.es; 4Department of Medical Biochemistry, Molecular Biology and Immunology, University of Seville, 41004 Seville, Spain

**Keywords:** cerium oxide nanoparticles, inflammation, reactive oxygen species, superoxide-dismutase, catalase, regeneration, tissue engineering

## Abstract

Cerium oxide nanoparticles (CNPs) are biocompatible nanozymes exerting multifunctional biomimetic activities, including superoxide dismutase (SOD), catalase, glutathione peroxidase, photolyase, and phosphatase. SOD- and catalase-mimesis depend on Ce^3+^/Ce^4+^ redox switch on nanoparticle surface, which allows scavenging the most noxious reactive oxygen species in a self-regenerating, energy-free manner. As oxidative stress plays pivotal roles in the pathogenesis of inflammatory disorders, CNPs have recently attracted attention as potential anti-inflammatory agents. A careful survey of the literature reveals that CNPs, alone or as constituents of implants and scaffolds, strongly contrast chronic inflammation (including neurodegenerative and autoimmune diseases, liver steatosis, gastrointestinal disorders), infections, and trauma, thereby ameliorating/restoring organ function. By general consensus, CNPs inhibit inflammation cues while boosting the pro-resolving anti-inflammatory signaling pathways. The mechanism of CNPs’ anti-inflammatory effects has hardly been investigated, being rather deductively attributed to CNP-induced ROS scavenging. However, CNPs are multi-functional nanozymes that exert additional bioactivities independent from the Ce^3+^/Ce^4+^ redox switch, such as phosphatase activity, which could conceivably mediate some of the anti-inflammatory effects reported, suggesting that CNPs fight inflammation *via* pleiotropic actions. Since CNP anti-inflammatory activity is potentially a pharmacological breakthrough, it is important to precisely attribute the described effects to one or another of their nanozyme functions, thus achieving therapeutic credibility.

## 1. Introduction

Inflammation is a complex, multifunctional defensive response set up by the organism in response to harmful stimuli, such as pathogens, irritants, external injuries, damaged cells, etc. It involves the cooperation of multiple players, including innate immune cells (e.g., neutrophils and macrophages), adaptive immune cells (e.g., T cells), and endothelial cells, to eliminate the initial inflammatory stimulus and trigger tissue repair. The main molecular mediators and regulators of inflammation are cytokines, chemokines, eicosanoids, and reactive oxygen species (ROS) [1,2,3]. Though there is a safe-guard system, often inflammation deranges and becomes chronic, being a leading cause of organ malfunctioning, chronic pathologies, and even death. Anti-inflammatory measures are thus an important goal of clinical and pharmacological research. However, due to the complexity of the process, resolutive therapeutic strategies (i.e., that are not limited to the symptomatic treatment) are still missing and require urgent solutions.

The main targets of anti-inflammatory therapies are presently ROS. These highly reactive molecules are partially reduced metabolites of oxygen, including superoxide anion (O_2_^•-^), singlet oxygen (^1^O_2_), hydroxyl radical (^•^OH), and hydrogen peroxide (H_2_O_2_), that can oxidize many biomolecules, exerting regulative or toxic effects depending on their levels. ROS can be generated as byproducts of cellular metabolism, e.g., the electron transport chain in mitochondria, or are actively produced by pro-oxidant enzymes such as oxygenases, cytochrome P450, or the NADPH oxidases [1]. At low, physiological concentrations, they serve as key signaling molecules modulating cell growth, differentiation, senescence, and apoptosis [1,4,5]. At higher concentrations they exert noxious effects, damaging proteins, lipids, and nucleic acids [1,6], subverting intra- and extra-cellular communication. Therefore, aerobic organisms are equipped with endogenous antioxidant systems, including the enzymes superoxide dismutase (SOD), which dismutates O_2_^•-^ to H_2_O_2_, and catalase, which converts 2H_2_O_2_ to 2H_2_O + O_2_, reducing step-by-step molecular O_2_ performing efficacious scavenging actions. Moreover, the cellular glutathione (GSH) system [7] allows additional ROS scavenging *via* the GSH redox cycle, a process in which GSH is cyclically oxidized to GSSG by glutathione peroxidase (GPx), while reducing ROS, and reduced back by GSH-reductase. Additionally, non-enzymatic defenses comprise exogenous (i.e., diet-provided), non-recyclable molecules such as all trans retinol 2 (vitamin A), ascorbic acid (vitamin C), α-tocopherol (vitamin E), and β-carotene.

When the balance between ROS and antioxidants is altered in favor of oxidations, cells undergo oxidative stress, a condition facilitating or even causing most human pathological conditions [8,9,10].

This suggested that external administration of exogenous antioxidants (“antioxidant therapy”) might revert pathological conditions. Unfortunately, most antioxidants are poorly bioavailable and they mostly cannot be regenerated when oxidized, thus requiring high doses and continuous assimilation [11,12]. In fact, no efficacious antioxidant therapy has been devised so far and the term remains essentially a conceptual issue.

Some inorganic nanoparticles (i.e., materials acquiring peculiar activities when in the 1–100 nm size range), including fullerenes, Pt nanoparticles, gold nanoparticles, and cerium oxide nanoparticles (CNPs or nanoceria), possess intrinsic antioxidant properties [13]. Although the activity of the former nanoparticles is ill-defined, the mechanism through which CNPs scavenge the most noxious ROS has been thoroughly studied; therefore, CNPs are receiving much attention in the field of nanomedicine [13]. Cerium is a lanthanide rare earth element possessing Ce^4+^ and Ce^3+^ oxidation states. When in nanosized particles, the two oxidation states co-exist on the nanoparticle surface, determining a dynamic redox switch [14]. This allows eliminating ROS and reactive nitrogen species (RNS) through a self-regenerating redox cycle: CNPs perform a SOD mimetic activity when Ce^3+^ is oxidized to Ce^4+^ by reducing superoxide or peroxynitrite to peroxide or nitrate, respectively [15,16], and a catalase-mimetic activity when Ce^4+^ is reduced to Ce^3+^ by the oxidation of hydrogen peroxide to molecular oxygen [17]. CNPs are thus nanozymes able to undergo a complete energy-free redox cycle, cyclically scavenging the most noxious free radicals, sharing the recycling ability of endogenous intracellular antioxidant defenses but without the energy required for the recycling of the latter (e.g., glycolysis as required for the GSH redox cycle, ATP, NADH, etc.). The long-lasting antioxidant effect of CNPs is not only observed in chemical reactions, but has been experimentally demonstrated in biological systems, showing that a single administration provides antioxidant protection even after many cell division cycles [18]. This, coupled to the well-evident biocompatibility shown by the numerous studies performed *in vitro* and *in vivo* [19,20,21], has created a mounting interest in CNPs as unprecedented antioxidants [13,14], foreseeing an efficacious, unconventional antioxidant therapy.

Prolonged or chronic exposure to oxidative stress is a key event in the pathogenesis of inflammatory pathologies, such as autoimmune disorders, inflammatory bowel diseases, atherosclerosis, diabetes, and neurodegenerative diseases [1,7,22]. An antioxidant-based therapeutic approach would therefore be ideal for the treatment of inflammation-related pathologies. However, as stressed above, the low efficacy of standard molecular antioxidants have so far limited their development. The long-lasting antioxidant effect of CNPs therefore provides hope for the development of efficacious anti-inflammatory therapy: indeed, many *in vivo* studies report encouraging, at times even surprising, results.

Considering the growing importance of the issue related to the anti-inflammatory effects of CNPs, we review and comment here on the abundant literature dealing with the anti-inflammatory potential of CNPs. We pay particular attention to the analysis of the mechanism(s) through which CNPs interfere with the inflammatory response, exploring whether current data allow us to attribute the anti-inflammatory effects of CNPs specifically to ROS scavenging (i.e., the Ce^4+^/Ce^3+^ redox switch) or whether any other of the multiple catalytic activities of CNPs may contribute to the final effects.

## 2. Anti-Inflammatory Application of Cerium Oxide Nanoparticles

Chronic inflammation is involved in most physical and mental disorders and is now recognized as the major cause of death in the world [23]. ROS play a key role in the pathogenesis of inflammatory pathologies; hence, CNP-mediated ROS scavenging represents a promising therapeutic approach to attenuate inflammation. In 2009, Hirst *et al*. were among the first to propose CNPs as potential breakthrough therapeutic agents against inflammation, showing that CNPs suppressed lipopolysaccharide (LPS)-induced oxidative stress and inflammation in murine macrophages [24] and displayed optimal tolerability *in vivo* (C57BL/6 mice). Since then, *in vitro* and *in vivo* reports demonstrating that CNPs may strongly ameliorate a wide range of inflammatory-based pathologies have multiplied.

The following sections collect the most recent data on CNP application in neurodegenerative and autoimmune diseases, liver inflammation, gastrointestinal disorders, and ocular pathologies, as well as bacterial infections and traumatic injuries.

### 2.1. Neurodegenerative Diseases

It is well-established that prolonged oxidative stress is involved in the pathogenesis of several neurodegenerative conditions, including Alzheimer’s disease, a degenerative disorder of the central nervous system (CNS) that causes mental deterioration and progressive dementia [25]. The main pathological indicator of this disease is the accumulation of amyloid-beta (Aβ) plaques, which can cause severe mitochondria dysfunction leading to neuron death by excessive ROS production [25,26]. The cytoprotective effects of CNPs on neuronal cells was explored by D’Angelo *et al*. [27], who treated differentiated SH-SY5Y neuroblastoma cells with Aβ (fragments 25–35) ± CNPs. The nanoparticles protected the culture from Aβ-induced cell death and neurite atrophy, and preserved cytoskeletal organization: the mechanism implies CNP-induced stimulation of brain-derived neurotrophic factor (BDNF) and its high-affinity receptor TrkB, at both the mRNA and protein levels.

The efficacy of CNPs against Alzheimer’s disease was investigated *in vivo* by Kwon and colleagues in a 5XFAD transgenic Alzheimer’s disease mouse model [28]. The authors prepared triphenylphosphonium (TPP)-conjugated CNPs to target nanoparticles to mitochondria upon internalization, and found that TPP-CNPs mitigated oxidative stress and suppressed neuronal death. The authors also showed that TPP-CNPs prevented activation of microglia, which are resident CNS immune cells that polarize to pro-inflammatory phenotypes during AD progression, leading to neuroinflammation. The ability of CNPs to mitigate microglia-induced inflammation was attributed to the Ce^3+^/Ce^4+^ redox switch due to mitochondrial localization; however, further analyses were not performed.

Jia *et al*. also explored the effect of CNPs on LPS-stimulated BV2 microglial cells [29]. The authors administered CNPs engineered with LXW7, a small cyclic peptide able to bind integrin αvβ3 and block its downstream signaling, thus impeding microglia activation. The results showed that CNPs efficiently reduced ROS production and potentiated the anti-inflammatory activity of the peptide, abating the expression of pro-inflammatory cytokines, such as interleukin (IL)-1β and tumor necrosis factor alpha (TNF-α). CNPs also reduced phosphorylation (i.e., activation) of the key mediators of microglia activation FAK and STAT3. Although the biological mechanism was thoroughly explored, the intrinsic mechanism through which CNPs blunt inflammation was not elucidated at the material level.

In a more recent work, Machhi *et al*. showed that europium-doped CNPs reduced the production of pro-inflammatory IL-6 and IL-1β and counteracted the expression of inducible nitric oxide synthase (iNOS) in LPS-treated BV2 microglia cells [30]. Of note, CNP treatment also promoted uptake and metabolism of Aβ fragments by BV2 cells, thus highlighting that CNPs not only attenuated microglia-induced inflammatory burden but also restored the homeostatic phenotype of microglial cells, favoring the correct elimination of Aβ plaques.

Along with microglia, astrocytes are key glial cells in the CNS contributing to neuronal homeostasis, maintenance of the blood–brain barrier, and synaptic functioning [31]. In 2016, Xu *et al*. explored CNP efficacy against neurodegeneration induced by long-term exposure to particulate matter smaller than 2.5 μm (PM2.5) [32]. CNP administration to PM2.5-stimulated astrocytes strongly reduced the release of pro-inflammatory cytokines, such as IL-6, IL-1β, and TNF-α. CNPs transactivated expression of antioxidant genes, including SOD1 and SOD2, and the authors attributed CNP anti-inflammatory action to SOD overexpression. However, the mechanism by which CNPs increased SOD expression is unclear considering that antioxidants generally reduce, rather than increase, the expression/activity of antioxidant enzymes, due to their ability of providing a surplus of antioxidant potential.

CNPs have also exhibited neuroprotection in amyotrophic lateral sclerosis (ALS) [33], a rare neurological disease that affects motor neurons, causing progressive loss of muscle control. This was investigated by De Coteau *et al*. in SOD1^G93A^ transgenic mice, a recognized *in vivo* model of ALS. The authors reported that bi-weekly intravenous tail administration of CNPs markedly ameliorated the defective motor function and longevity; this therapeutic effect was associated with the antioxidant activity of CNPs, though no *ad hoc* investigation is described.

The loss of neurons and impaired synaptic plasticity are not only hallmarks of neurodegenerative diseases but also of major depressive disorders and traumatic brain injuries. In the first case, the oxidative and nitrative stress caused by chronic psychosocial or physical stress may lead to a decrease in neurogenesis and to the loss of neurotrophic support and synaptic plasticity in the hippocampus [34]. In a recent work, Zavvari and colleagues reported the neuroprotective activity of CNPs in experimental stress-induced depression in male rats [35]. The authors showed that a single dose of CNPs significantly reduced rats’ depressive-like behavior, attenuated oxidative stress (as shown by reduced malondialdehyde, MDA) and inflammation (reduced IL-6), and promoted hippocampal neurogenesis and synaptogenesis. CNP antioxidant actions were also in this case hypothesized as the mechanism involved.

Youn *et al*. recently investigated CNPs in a mouse model of mild traumatic brain injury [36]. The authors showed that retro-orbital injection of CNPs was effective in preventing neuronal cell death, brain edema, and down-regulating expression of SOD1, SOD2, and cyclooxygenase-2 (COX-2), which is a major pro-inflammatory enzyme converting arachidonic acid into prostaglandins. This led to a strong improvement in mouse cognitive function.

Table 1 summarizes the main informative data from the articles cited in this first section. 

### 2.2. Inflammatory Autoimmune Diseases

Inflammatory autoimmune diseases imply the anomalous response of the immune system against healthy tissues and organs.

Rheumatoid arthritis is a chronic autoimmune disease characterized by the inflammation of multiple joints, resulting in the progressive disruption of cartilage and bones [37]. It is associated with elevated levels of ROS, which promote lipid peroxidation, protein oxidation, DNA damage, and activate nuclear factor kappaB (NF-κB), which transcribes pro-inflammatory genes. This leads to macrophage polarization towards the pro-inflammatory M1 phenotype, which, in a feed-forward loop, contributes to augmenting the levels of ROS and pro-inflammatory cytokines [38]. In 2020, Kalashnikova *et al*. synthetized redox-active albumin-conjugated CNPs (A-CNPs) and evaluated the anti-inflammatory efficacy of the composite in a mouse model of collagen-induced arthritis [39]. The authors reported that A-CNPs-treated mice presented fewer infiltrated M1 and more M2 macrophages, suggesting that A-CNPs (re-)addressed macrophages towards the M2 anti-inflammatory phenotype. This resulted in a decrease in the disease severity to an extent comparable to methotrexate, the standard antirheumatic treatment (which has severe side effects), proposing CNPs as a less toxic potential alternative.

The efficacy of CNPs in reducing inflammatory immune cell infiltration in tissues was also reported by Kim *et al*. [40]. They synthetized citric-acid-coated CNPs (CA-CNPs) displaying SOD/catalase-mimetic activity and tested them in a mouse model of acute hind paw inflammation. CA-CNPs significantly reduced immune cell infiltration, lowering the levels of pro-inflammatory cytokines TNF-α and IL-1β, and reducing paw edema and pain hypersensitivity.

Rheumatoid arthritis strongly favors the onset of osteoarthritis, which leads to abnormal bone remodeling and progressive cartilage degeneration [41]. In 2020, Lin *et al*. showed that CNP administration to chondrocytes treated with H_2_O_2_, to mimic the pro-oxidant environment of osteoarthritis, increased the therapeutic efficacy of hyaluronic acid, a major non-surgical option for the treatment of osteoarthritis [42]. In particular, CNPs promoted the expression of genes involved in collagen (COL1A1 and COL2A1) and aggrecan (ACAN) deposition, phenomena that are strongly impaired in osteoarthritis.

The efficacy of CNPs against osteoarthritis was later confirmed in an elegant work by Xiong *et al*. [43]. The authors explored the protective effects of CNPs on chondrocytes and cartilage explants exposed to IL-1β to mimic *in vitro* the pro-inflammatory microenvironment typical of osteoarthritis. CNPs reduced the levels of ROS and NO, and the expression of pro-inflammatory iNOS, IL-6, and cyclooxygenase-2 (COX2), while significantly increasing the expression of Nrf2, a key nuclear factor regulating the expression of genes involved in the cellular oxidative defense system. These effects ended in reduced inflammation, decreased extracellular matrix (ECM) degradation, and increased cell survival. To explore whether the protective effects exerted by CNPs were mediated by Nrf2 activation, the authors established stable Nrf2-deficient chondrocytes and exposed them to IL-1β and CNPs. The antioxidant and cytoprotective effects of CNPs on the ECM were almost inhibited, indicating that CNPs could protect chondrocytes from oxidative stress by activating the Nrf2/HO-1 signaling pathway

Multiple sclerosis (MS), an inflammatory autoimmune disease, is a debilitating neurological disorder in which T cells infiltrate the brain and spinal cord parenchyma and target myelin sheath, inducing local inflammation that involves the activation of microglia, astrocytes, and blood-derived macrophages, causing white matter lesions [44]. Eitan *et al*. first evaluated the therapeutic efficacy of CNPs + lenalidomide, an immune-modulating drug, in the experimental autoimmune encephalomyelitis (EAE) mouse, which is a standard murine model of human MS [45]. The authors showed that the combined treatment enhanced lenalidomide performance, reducing myelin loss, mitigating inflammation, and reducing TNF-α and interferon (IFN)-γ levels. This resulted in a near elimination of encephalomyelitis symptoms.

Table 2 reports the main informative data from the articles cited in this section dedicated to autoimmune diseases.

### 2.3. Liver Inflammation

Inflammation is one of the most characteristic features of chronic liver diseases, regardless of their origin (alcoholic, fatty, autoimmune, viral, etc.) [46]. Non-alcoholic fatty liver disease (NAFLD), for example, begins with a phase of hypertriglyceridemia with abnormal hepatic lipid metabolism, followed by a second phase of inflammation and oxidative stress [47]. In 2017, Kobyliak and colleagues investigated the anti-inflammatory potential of CNPs in a rat model of NAFLD associated with monosodium glutamate (MSG)-induced obesity [48]. The intermittent administration of CNPs was demonstrated to prevent liver damage and reduce lobular inflammation, the content of triglycerides and total liver lipids, mice body weight, and visceral adipose tissue mass. Moreover, CNPs decreased the serum levels of pro-inflammatory IL-1β and IL-12Bp40, while increasing the levels of anti-inflammatory IL-4 and TGF-β. The authors hypothesized that the anti-inflammatory effect could depend on the strong antioxidant activity of CNPs but state that the precise mechanism of action remains to be revealed.

In 2021, Abbasi *et al*. further confirmed the protective effects of CNPs in NAFLD in Wistar rats [49]. CNP administration attenuated liver oxidative stress (as shown by decreased MDA and increased GSH) and inflammation (reduced TNF-α), thus strongly alleviating pathological liver damage. A more recent work by Lebda *et al*. demonstrated the anti-steatosis action of CNPs in a rat model of postmenopausal obesity [50]. The authors showed that CNPs reduced the serum level of MDA and the hepatic level of TNF-α and lowered the infiltration rate of inflammatory cells, mitigating the liver steatosis, steatohepatitis, and fibrosis of obese rats. This was accompanied by signs of improved liver functions, such as decreased rat visceral obesity, reduced serum levels of aspartate transaminase (AST), alanine transaminase (ALT), triglycerides, total and low-density lipoprotein (LDL) cholesterol, and increased the level of high-density lipoprotein (HDL) cholesterol.

NAFLD, hepatitis, and non-alcoholic steatohepatitis may lead to liver fibrosis, a homeostatic disturbance characterized by the loss of epithelial features in hepatocytes in favor of mesenchymal trans-differentiation, and accumulation of extracellular matrix (ECM) proteins, such as collagen, which disrupts the hepatic architecture forming a fibrous scar. Advanced liver fibrosis eventually leads to cirrhosis, portal hypertension, and liver failure [51]. In 2016, Oró and colleagues explored the hepatoprotective properties of CNPs in rats bearing liver fibrosis [52]. Fibrosis was induced in rats by exposing them to carbon tetrachloride (CCl_4_) vapor, a well-known hepatotoxin that in the long run may lead to organ fibrosis. CNPs administration to CCl_4_-treated rats strongly decreased the number of infiltrating monocytes/macrophages (CD68-positive cells) and the expression of the pro-inflammatory molecules IL-1β, TNF-α, iNOS, and COX-2 in liver tissue, significantly improving liver functions. The authors investigated the ROS-scavenging action of CNPs by quantitative RT-PCR on rat liver tissues, finding that CNPs reduced mRNA expression of neutrophil cytosol factors 1 and 2 (oxidases involved in superoxide anion production) and eosinophil peroxidase, which catalyze the formation of hypohalous acids (potent oxidizing agents) from hydrogen peroxide and halide ions in solution.

More recently, Godugu *et al*. confirmed the ability of CNPs to alleviate liver fibrosis in a bile duct ligation (BDL) mice model [53]. BDL-operated mice were subjected to intraperitoneal administration of CNPs, which decreased oxidative stress (nitrite and MDA levels), infiltration of inflammatory cells into liver tissue, levels of pro-inflammatory cytokines (IL-1β, IL-6, and TNF-α) and enzymes (COX-2 and iNOS), and, consequently, reduced the levels of pro-fibrotic TGF-β1, Snail, TIMP1, LOXL-2, N-cadherin, fibronectin, and Collagen I.

Hepatic failure may require liver transplantation, which unfortunately can have complications; among the worst is hepatic ischemia reperfusion (IR), which often leads to graft failure [51,54]. Hepatic IR is a biphasic phenomenon consisting of a temporary interruption in blood supply to the liver, followed by sudden reperfusion that disrupts liver homeostasis, resulting in free radical production and Kupffer cell (resident liver macrophages) activation, triggering robust inflammation [54,55]. Manne *et al*. first reported that prophylactic treatment with CNPs attenuated hepatic IR injury in Sprague-Dawley rats [54]. CNP administration 20 min before the hepatic IR injury reduced serum ALT and lactate dehydrogenase (LDH) levels, preserved normal hepatocellular architecture, and lowered the serum levels of inflammatory markers. A later work by Zengin *et al.* delved into the biochemical parameters regulated by CNPs [56]. Nanoparticle administration 24 h before the hepatic IR injury, protected mice from liver damage (reduced serum LDH, ALT, and AST), prevented tissue lipid peroxidation (reduced MDA and increased GSH and the GSH/GSSG ratio), and lowered the plasma level of matrix metalloproteinases (MMP-2 and MMP-9) and pro-inflammatory cytokines (TNF-α, IL-1α, IL-1β, IL-2, IL-6, IL-8, and IL-12). The anti-inflammatory cytokine IL-10 was enhanced by CNPs.

Table 3 summarizes the main data from the articles cited in this section.

### 2.4. Gastrointestinal Inflammatory Disorders

Peptic ulcer disease (gastric and duodenal) is a multi-causal disorder characterized by disruption of the inner lining of the gastrointestinal tract due to gastric acid secretion or pepsin [57]. It occurs when the balance in the gastrointestinal tract between the biological defenses and aggressive external factors, including nonsteroidal anti-inflammatory drugs (NSAIDs), *Helicobacter (H.) pylori*, stress, etc., is disturbed [57,58].

In 2013, Prasad *et al*. first reported the protective effect of CNPs against ethanol-induced gastric ulcers in Sprague-Dawley rats [59]. A more recent work also showed the efficacy of CNPs against stress-induced gastric mucosa lesions in rats [60]. In particular, the prophylactic administration of CNPs reduced the ulcer area by 42%, protecting collagen from degradation, reducing oxidative stress (in terms of lipid peroxidation and H_2_O_2_ content), and mitigating inflammation (decreased level of IL-1β and IFN-γ and increased IL-4, IL-10, and TGF-β). The authors hypothesized, from the literature data, that the anti-inflammatory action could be mediated by a CNP-induced redox modulation of NF-κB; however, the exact molecular mechanism was not investigated. 

Recently, Asgharzadeh *et al*. explored the potential of CNPs as therapeutic agents for the treatment of ulcerative colitis [61], one of the most diagnosed inflammatory bowel diseases causing mucosal damage and crypt loss. The authors functionalized CNPs with sulfasalazine (SSZ), a first-line drug for treating ulcerative colitis patients and administered the conjugate to C57BL/6 mice treated with dextran sodium sulfate, an established model of mice colitis. SSZ-CNPs ameliorated colitis clinical symptoms better than SSZ alone and attenuated colon tissue damage, in terms of both mucosal damage and crypt loss. The effect was attributed to CNP-induced reduction of lipid peroxidation and increase in antioxidative factors, including total thiols level, SOD, and catalase activities. Also in this case, the reason why antioxidant agents such as CNPs would potentiate the endogenous antioxidant defenses was not explained.

Table 4 summarizes the main data from the articles cited in this section.

It must be noted that CNPs at low pH values (<4.5), such as when they are in gastric fluids, may dissolve and release toxic Ce^3+^ ions [62,63]. Theoretically, this would hinder, or even cancel, the therapeutic benefits of CNPs against gastric disorders. However, recent studies [64,65] have shown that CNPs did not significantly dissolve in simulated gastric fluid nor lose their biological activity [65], possibly because of fluid components protecting the nanoparticles from dissolution [66]. This may provide a logical explanation for the success of the *in vivo* studies described above.

### 2.5. Ocular Inflammation

Oxidative stress plays a key role in the development of inflammation-related ocular pathologies, including age-related macular degeneration [67], neovascular diseases [68], glaucoma [69], and retinopathies [70]. The retina, for example, is highly vulnerable to oxidative stress due to its high consumption of oxygen and exposure to light [70]. In 2006, Chen *et al*. reported that CNPs reduced the intracellular level of ROS in primary cell cultures of rat retina and prevented loss of vision due to light-induced degeneration of photoreceptor cells *in vivo* [71]. In a more recent study, Kyosseva *et al*. demonstrated that CNPs inhibited the expression of genes associated with inflammation and angiogenesis in a mouse model of retinal angiomatous proliferation [72]. In particular, a single intravitreal injection of CNPs in both eyes allowed the reduction of pro-inflammatory cytokines and pro-angiogenic factors, by inhibiting the phosphorylation, and thus the activation, of MAPKs (including pERK1/2, pJNK1/2, and p38) and AKT.

More recently, Zheng *et al*. explored the cytoprotective effect of CNPs on corneal neovascularization [73]. The authors first investigated CNP antioxidant and anti-inflammatory activities *in vitro* in human corneal epithelial cells and murine RAW 264.7, respectively. Next, they tested CNPs in a rat model of inflammation-associated corneal neovascularization, reporting that CNPs reduced the release of the pro-inflammatory cytokine TNF-α, significantly attenuating corneal vascularization and opacification. Regarding the mechanism of action, the authors hypothesized that CNPs may lower inflammation through the down-regulation of NF-κB mediated by CNP-induced ROS suppression.

More recently, Badia *et al*. developed a formulation of CNPs for ocular topical administration [74] that was tested *in vitro* and *in vivo* for efficacy against macular degeneration. *In vitro*, the formulation was biocompatible and showed antioxidant activity, reducing oxidative stress in retinal ARPE19 cells. *In vivo*, using the macular degeneration DKOrd8 mouse model, the formulation could reach the retina after topical delivery, attenuating inflammation and normalizing the altered retinal transcriptome of DKOrd8, effectively delaying macular degeneration.

Table 5 summarizes the main data from the articles cited in this section.

### 2.6. Pathogen-Induced Inflammation

Invasion by foreign pathogens sets in motion the immune response, which includes ROS production by polymorphonuclear neutrophils (PMNs), a key event in the killing and clearance of pathogens. However, excessive ROS production may lead to tissue injury [1]. Serebrovska *et al*. explored the antioxidant and anti-inflammatory effect of silica nanoparticles with immobilized CNPs on their surfaces (S-CNPs) in rat experimental pneumonia [75]. S-CNPs reduced the level of ROS in whole blood and lung tissue, and the expression of pro-inflammatory cytokines (TNF-α and IL-6) and chemokines (CXCL2), leading to reduced lung tissue injury. In addition, S-CNPs stimulated lung functions, measured as oxygen consumption, in both healthy and pneumoniae rats. The authors, though aware that a precise mechanism was not demonstrated, hypothesize that it consists in CNP-mediated ROS scavenging.

A dysregulated host response to infections may lead to sepsis, a serious life-threatening medical emergency eventually resulting in multiple organ dysfunction and failure [76]. Sepsis is often accompanied by the development of a non-specific systemic inflammatory response syndrome (SIRS) that is characterized by cytokine storm [77] and elevated ROS levels produced by monocytes and macrophages. If not controlled, SIRS can lead to septic shock and death [78].

In addition to the anti-inflammatory potential, CNPs also exhibit direct antibacterial action [78,79], which was recently attributed to CNP haloperoxidase activity [80,81,82], i.e., the ability to mediate the oxidation of halides (bacterial communication molecules) by H_2_O_2_ and thereby hamper bacterial communications. The dual activity of CNPs to inhibit microbial growth on the one hand and reduce inflammation on the other, make them promising agents to treat severe sepsis.

Here follows a series of reports describing the ameliorative effect of CNPs on sepsis-induced damage in different organs. Selvaraj *et al*. investigated CNP effects in infectious cecal peritonitis in Sprague-Dawley rats [78]. CNP administration decreased the serum levels of IL-6 and blood urea nitrogen, and prevented sepsis-induced histopathological changes in the lungs; this led to increased animal survival from 20 to 90%. To explore the potential mechanism of action, the authors moved to an *in vitro* system of LPS-stimulated murine RAW 264.7 macrophages; here, CNPs reduced the levels of ROS, TNF-α, IL-1β, and IL-6 and nearly abrogated LPS-induced nuclear translocation of NF-κB/p65. The authors also investigated the protective effect of CNPs against LPS-induced hepatic dysfunction in Sprague-Dawley rats [83]. In this model, CNPs attenuated inflammation by reducing serum levels of TNF-α, IL-1β, and IL-1α, the expression of phosphorylated MAPKs (such as p-38 and ERK1/2), the infiltration rate of immune cells into the portal area, and the levels of bilirubin and ALT. This led to improved liver homeostasis in the rats and dramatically reduced mortality from 70 to 10%.

More recently, Rice *et al*. reported that a single intravenous administration of CNPs attenuated sepsis-induced splenic damage, pro-inflammatory cytokine release, and bacterial load in the blood and peritoneal fluid of Sprague Dawley rats [84].

In 2015, Manne and colleagues tried to deepen the understanding of mechanisms underlying CNPs’ antiseptic activity in polymicrobial sepsis Sprague-Dawley rats [85]. First, they showed that tail intravenous administration of CNPs attenuated hypothermia and reduced the serum levels of ROS and pro-inflammatory cytokines, totally preventing rat mortality (at least up to 72 h). Then, they explored the effect of CNPs on mitogen-activated protein kinases (MAPKs), NF-κB, and Stat signaling, whose activation is associated with the peritonitis-induced inflammatory cascade [86]. CNP treatment was found to significantly decrease the phosphorylation of ERK 1/2 and Stat-3 at 18 h post-treatment, thus impairing the pro-inflammatory signaling. CNP-mediated dephosphorylation of Stat-3 in a rat model was also reported by Asano *et al*. [87].

The therapeutic performance of CNPs against sepsis was later improved by Jeong *et al*. in a SIRS mouse model [88]. The authors synthetized 6-aminohexanoic-acid-conjugated CNPs (6-AHA-CNPs) to increase the dispersibility and ROS scavenging potential of CNPs in aqueous media. This increased *in vitro* efficacy of CNPs to reduce ROS and IL-1β production in LPS-stimulated RAW 264.7 macrophages; in *in vivo* experiment, the increased dispersibility of 6-AHA-CNPs increased CNP performance in reducing pulmonary interstitial edema, vascular congestion, and inflammatory cell infiltration in the lungs and liver, strongly increasing the survival rate of SIRS mice.

Table 6 summarizes the main data from the articles cited in this section.

### 2.7. Inflammation in Tissue-Engineering-Treated Traumatic Injuries: Premises and Promises of CNPs

Many traumatic injuries may not heal on their own and, in severe cases, they may result in permanent disability [89]. Tissue engineering (TE) can help in such situations, for example, by manufacturing stem-cell-based tissues to be implanted into damaged organs to restore or improve their function. The success of TE depends on many factors, including the extent of the lesion, the type of organ that has been damaged, and the inflammatory response of the host. As for the latter, the cellular components of the engineered tissue (i.e., the stem cells that should differentiate and proliferate to reconstitute the damaged tissue) might suffer/die in the inflammatory microenvironment setup by the damaged tissue, frustrating engraftment and leading to implant rejection. Among the immune cells involved in injury-mediated inflammation, macrophages play a central role, being at the frontier between innate and adaptive immunity. In particular, M1 pro-inflammatory macrophages are considered an obstacle to regeneration, whereas the anti-inflammatory M2 macrophages play a pro-regenerative role. Modulation of macrophage plasticity towards the M2 phenotype therefore represents a key strategy in the success of TE for regenerative medicine [90,91].

A general trend emerges on the basis of the anti-inflammatory activities performed by CNPs and discussed in this review, which is the reduction of pro-inflammatory and potentiation of the anti-inflammatory response, a phenomenon associated with (and a marker of) macrophage polarization towards the anti-inflammatory M2 phenotype, an event molecularly identified in [39]. Therefore, the contribution that CNPs can make to TE applications consists of helping scaffolds and implants to modulate macrophage polarization and favor of tissue regeneration [91,92,93].

Among severe traumatic injuries, traumatic spinal cord injury (SCI) is one of the most devastating. It may cause permanent motor and sensory dysfunction [94]. After the primary accidental trauma, a secondary self-inflicted lesion may follow, involving severe oxidative stress and inflammation, which may lead to cytotoxic nerve excitation and neuron apoptosis, thus preventing healing [95]. Kim *et al*. showed that CNPs [96] dose-dependently allow functional recovery of experimentally contused spinal cord in rats after a single injection at 0.5 and 1 mg/kg, significantly reducing the number of iNOS-positive cells, infiltration of inflammatory cells, lesion cavity size, and improving rat locomotor functions. To investigate the molecular mechanism, the authors analyzed the expression of key genes involved in ROS generation and inflammation, reporting that CNP administration down-regulated iNOS, COX-2, Nr-f2, p53, caspase-3, IL-1β, IL-6, TNF-α, and leukemia inhibitory factor levels.

A promising cell-based approach to promote SCI repair involves the use of neural stem cells (NSCs) [97]; however, its efficacy is limited by the oxidative and pro-inflammatory microenvironment that arises after SCI, which promotes NSC death. To overcome such limitations, in a recent study performed in a spinal cord transection rat model by Liu *et al*., NSCs were accompanied by CNP-based hydrogels, with the goal of protecting NSCs and enhance nerve repair [98]. Experimentally, albumin-conjugated CNPs were dispersed in gelatin methacryloyl to obtain a hydrogel with robust ROS scavenging potential, within which NSCs were loaded. The CNP gel improved the survival rate of NSCs and promoted the infiltration of endogenous cells, resulting in improved rat motor function. Regarding inflammation, the CNP gel attenuated ROS and lipid peroxidation, reduced the infiltration of inflammatory cells, and polarized microglia towards the M2 phenotype, thus creating a pro-resolutive environment favoring regeneration and remyelination.

Broken bones generally heal but, in many cases, healing is impossible, incomplete, or late; this constitutes an important problem in an aging society, thus requiring remedies. Indeed, the setup of TE-based implants for fractured bone healing is a hot topic in biomedical research. CNPs were shown to be very promising in this task. Wei *et al*. tested CNPs on LPS-stimulated macrophages and human bone mesenchymal stem cells (BMSCs) [99]. The authors showed that CNPs decreased M1-related pro-inflammatory cytokines (IL-1β and TNF-α) and iNOS levels, while enhancing the M2-related anti-inflammatory cytokines (IL-10 and TGF-β) levels. As far as the cellular components are concerned, BMSCs responded to CNP treatment by enhancing osteogenic differentiation, confirming the premise of using CNPs for TE applications. In 2016, Li *et al*. exploited CNPs to extend the biological performance of hydroxyapatite (HA)-based implants [100]. In particular, CNPs were incorporated into HA coatings *via* a plasma spraying technique. The addition of CNPs allowed polarizing murine RAW 264.7 macrophages towards the M2 phenotype. Indeed, CNPs decreased the expression of M1-associated surface receptors (CCR7 and CD11c) and pro-inflammatory cytokines, and enhanced the expression of M2-associated surface receptors (CD163 and CD206) and anti-inflammatory cytokines. Furthermore, CNPs promoted BMSC proliferation and osteogenic differentiation, increasing cell survival, mineral deposition, and the expression of osteogenic genes. In a more recent work, CNPs were applied to titanium substrate surfaces by magnetron sputtering and vacuum annealing [101]. *In vitro*, CNPs significantly enhanced the proliferation of rat BSMCs and the polarization of murine RAW 264.7 macrophages towards the M2 phenotype. *In vivo*, scaffolds were implanted into the intramedullary cavity of Sprague-Dawley rat femurs, significantly increasing bone volume and osteogenic gene expression.

Titanium implants are also commonly used for dental applications. One of the key factors contributing to the proper engraftment of dental implants is the prevention of bacterial adherence on the implant surface. In fact, if microbes adhere to the implant, they can cause peri-implantitis, an inflammatory lesion characterized by inflammation of the mucosa and loss of the supporting bone [102,103]. In 2019, Li *et al*. explored the potential of CNPs against peri-implantitis in *in vitro* and *in vivo* tests [103]. Titanium disks were modified with three different shapes of CNPs: rods, cubes, and octahedrons. The latter two shapes exhibited the best anti-bacterial activity *in vitro*, strongly reducing bacterial absorption on the disks. The inefficacy of rod-shaped CNPs was attributed to nanoscale rod walls, which might provide nucleation sites for protein absorption. When administered to LPS-stimulated macrophages, cube- and octahedron-shaped CNPs provided the strongest anti-inflammatory response, decreasing IL-6, TNF-α, and IL-1β levels and nuclear translocation of NF-κB/p65. When implanted into the backs of Wistar rats, octahedron-shaped CNPs had the best performance, attenuating the number of inflammatory cells infiltrating the tissue surrounding the implants and reducing the production of pro-inflammatory cytokines (IL-6, TNF-α, and IL-1β). The stronger anti-inflammatory activity exerted by octahedron-shaped CNPs was attributed to their smaller size and higher Ce^3+^/Ce^4+^ ratio.

The key role played by CNP-mediated regulation of NF-κB in controlling periodontal inflammation was later confirmed by Yu *et al*. [104]. The authors showed that CNPs reduced NF-κB activation *in vitro*, in LPS-stimulated macrophages, and *in vivo*, in a gingival inflammation rat model. Besides NF-κB, the authors also showed that CNPs reduced the phosphorylation of ERK1/2, p-38, and JNK MAP kinases. Unfortunately, the mechanism of action through which CNPs exerted such regulatory effects on the MAPK cascade was not investigated.

Table 7 summarizes the main data from the articles cited in this section.

## 3. CNPs: Nanotoxicology vs. Nanomedicine

Generally speaking, nanotoxicology and nanomedicine are not two sides of the same coin but totally different issues, since the former deals with the unwanted effects of involuntary environmental contact, whereas in the latter, a proper route for nanoparticle entry is carefully chosen to maximize the results and minimize adverse side effects. Nanotoxicology studies on the potential harm of nanoparticles as pollutants have covered a couple of decades, defining that they are harmful when inhaled, producing local and distal inflammation. Regarding alternative routes of penetration, such GI or dermal, the results are less clear; therefore, nanotoxicity seems to occur mainly through inhalation routes. This also applies to CNPs, which have been considered as a nanotoxicology issue due to concern for their massive use as diesel fuel catalysts. The consequences of CNP inhalation indeed showed pulmonary toxicity in *in vitro* and *in vivo* models [105,106,107,108], as well as exacerbation of pro-inflammatory conditions such as vascular injury in a rat model [109] and renal damage in cisplatin-treated rats (Nemmar *et al*., 2019 [110]). In such instances, CNPs act as pro-inflammatory agents, i.e., opposite to the sequela of data reviewed here. It is highly interesting to compare the Nemmar studies with that of Abdelhamid *et al*., who in 2020 demonstrated that CNPs administered by intraperitoneal injection ameliorated cisplatin- and oxaliplatin-induced renal toxicity in rats through anti-inflammatory activities [111], i.e., exactly the opposite of what happens upon CNP inhalation in practically the same model. This highlights how the route of entry influences the final result.

Of note, none of the nanomedicine-oriented articles considered in this review involved CNP administration *via* aerosol or intra-tracheal instillation, i.e., pulmonary exposure. The data reviewed and commented on here, in fact, highlights the biocompatibility of CNPs when using a proper exposure route, up to the point that their first use in humans was recently published [112].

## 4. Conclusions: Anti-Inflammatory = Antioxidant?

Here, we reviewed the abundant and coherent *in vitro* and *in vivo* literature data reporting CNPs’ anti-inflammatory effects, covering a wide range of clinically and socially relevant pathologies. These effects are generally attributed to the antioxidant activity of CNPs, which is a conceivable explanation since (a) oxidative stress plays a paramount role in inflammation and (b) the unique SOD- and catalase-mimetic action of CNPs seems enough *per se* to provide anti-inflammatory properties. However, CNPs are multifunctional nanozymes that perform additional redox and non-redox functions, including phosphatase, haloperoxidase, photolyase, and oxidase-like activities that occur independently from the catalase/SOD-mimetic activity [113] (see Figure 1). Such activities cannot be excluded *a priori* as being responsible for the anti-inflammatory effects reported, and *ad hoc* experiments should be performed to clarify the exact mechanism of action of CNPs as anti-inflammatory agents.

For example, the ability of CNPs to reduce the NF-κB and MAPK cascades, is generally attributed to CNP-mediated redox control of the inflamed microenvironment, even in the absence of experimental evidence [72,85]. However, this task could be conceivably attained through the phosphatase-like activity [114,115,116,117], i.e., with CNPs directly dephosphorylating MAPKs. This could also apply to NF-κB, whose activation, which is often oxidation dependent, also requires phosphorylation of its cytoplasmic anchor IκB [118], an event that may be directly prevented by the phosphatase activity of CNPs.

All these considerations point to CNPs as pleiotropic anti-inflammatory agents that combat inflammation from multiple sides and *via* multiple mechanisms, among which the antioxidant effect likely plays a major role. The many papers reviewed here, though thoroughly exploring the biological mechanisms, fail to investigate the mechanism at the material level, i.e., which catalytic function(s) of CNPs is/are required to prevent/fight inflammation. Indeed, several authors deduce (but not actually demonstrate) that it should consist in antioxidant action, whereas in most cases the authors plainly state that the mechanism still needs to be elucidated. Instead, when dealing with the bioactivity of CNPs, it should be experimentally demonstrated, or at least discussed, whether the anti-inflammatory effects reported are actually due to CNP redox switch. A clearly defined mechanism of action, in fact, would strengthen the credibility of the promising *in vitro* and *in vivo* data, providing molecular mechanisms that could be the turning point for predicting large-scale therapeutic applications of CNPs.

## 5. Future Directions

The therapeutic potential of CNPs as anti-inflammatory agents is impressive and deserves to be studied mechanistically in order to potentially develop breakthrough therapeutic strategies. Of note, their first use in humans against diabetic foot ulcers was recently published [112]. It would be therefore important to devise a strategy to experimentally demonstrate a cause–effect relationship between the Ce^3+^/Ce^4+^ redox switch and the anti-inflammatory effects of CNPs. In this view, to discriminate between the redox and non-redox activities of CNPs, Celardo *et al*. synthetized redox-inactive CNPs by doping them with 20% samarium (Sm) atoms, a lanthanide with fixed +3 valence, a modification that did not alter the original CNP mesh [19]. This tool allowed the authors to determine that the anti-apoptotic effect exerted by CNPs in drug-treated U937 leukemia monocytes was redox dependent [19], as was the cytoprotective effect on Jurkat cells against damage induced by UV exposure, a well-known pro-oxidant event [20]. On the contrary, it unveiled that CNPs’ radio-sensitizing effect on human immortalized HaCaT keratinocytes was independent from the redox switch, as this activity was shared by Sm-doped CNPs [119]. This implies that other, non-redox nanozyme-like activities of CNPs may have relevance in pathological contexts [120]. The use of such a tool could therefore be decisive to understand the mechanisms underlying the anti-inflammatory effect of CNPs.

## Figures and Tables

**Figure 1 nanomaterials-13-02803-f001:**
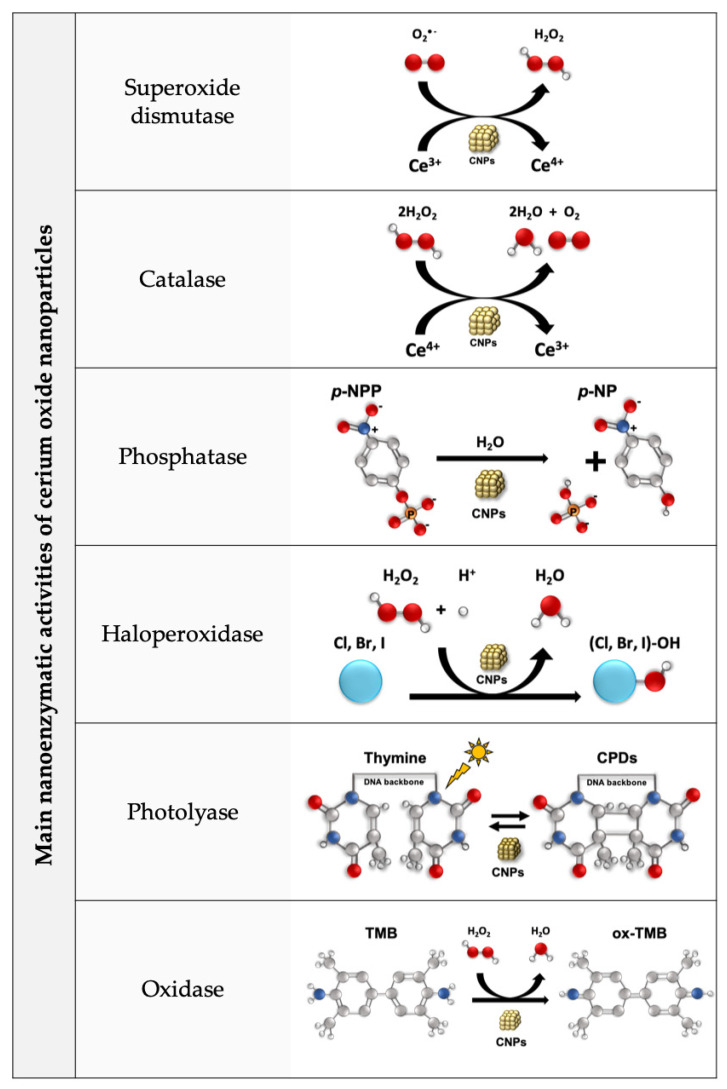
The multiple CNP enzymatic-like activities. Superoxide dismutase-like activity: CNPs reduce superoxide to hydrogen peroxide, oxidizing Ce^3+^ to Ce^4^. Catalase-like activity: CNPs oxidize hydrogen and molecular oxygen, reducing Ce^4+^ to Ce^3+^. The coupling of these two functions constitutes CNPs’ auto-regenerative redox cycle. Phosphatase-like activity: CNPs dephosphorylate para-nitrophenyl phosphate (*p*-NPP) in aqueous environments independently of the Ce^3+/4+^ redox switch and possibly due to oxygen vacancies [113]. Haloperoxidase-like activity: CNPs mediate the oxidation of halides (B, Cl, and I) by H_2_O_2_ into hypohalous acid (HOX) species. Photolyase-like activity: CNPs enable selective cleavage of dimers (e.g., cyclobutane pyrimidine dimers (CPDs)) into monomers. Oxidase-like activity: CNPs oxidase the colorimetric substrate 3,3′,5,5′-tetramethylbenzidine (TMB) to its oxidized form ox-TMB.

**Table 1 nanomaterials-13-02803-t001:** CNPs in neurodegenerative diseases: literature summary.

Nanomaterial	Synthesis Method	Morphology	Size (nm)		Model	Dosage	Route of Administration	Markers	Effects	Ref.
			Powder	Hydrodynamic Radius						
CNPs	Precipitation	Cubic	6–16	ND	SH-SY5Y neuroblastoma cells treated with 12.5 μM Aβ 25–35 for 24 h	100 μg/mL for 24 h	NA	↑ β-TubIII, GAP43, NF-H 200 ↑ GPX1, catalase ↓ SOD1, SOD2 ↑ BDNF, TrkB ↓ p-ERK1,2	↑ cell viability ↓ apoptosis ↑ antioxidant response ↓ neurite atrophy ↑ neuronal differentiation Maintenance of cytoskeletal organization	[27]
Triphenyl phosphonium-conjugated CNPs (TPP-CNPs)	Hydrolytic sol-gel	ND	ND	22	SH-SY5Y neuroblastoma cells and U373 astrocytoma cells treated with 5 μM Aβ	0.1 mM for 12 h	NA	↓mitochondrial ROS	↓ oxidative stress	[28]
					Transgenic 5XFAD mouse as *in vivo* model for Alzheimer’s disease (AD)	3 μL solution of 1 mg/mL	Unilateral subicular injection	↓ GFAP, Iba-1 ↓ 4-HNE	↑ neuronal viability ↓ glial cell activation ↓ oxidative stress	
CNPs modified with LXW7 peptide and polyacrylic acid (CeO_2_@PAA-LXW7)	EDC (1-ethyl-3-(3-dimethylaminopropyl carbodiimide) reaction	Spherical	ND	2–5	Murine BV2 microglial cells	1 μM	NA	↓ TNFα ↓ IL-1β ↓ ROS, NO ↓ p- FAK, p-STAT3	↓ inflammation	[29]
Europium doped CNPs (EuCNPs)	Solvothermal reactions	Spherical	ND	159.06	Murine BV2 microglial cells	100 ng/mL	NA	↑ CD36 ↑ co-localization of Aβ with Rab7 and LAMP1 ↓ IL-6, IL-1β	↑ microglial phagocytosis of Aβ ↓ inflammation ↓ Aβ plaques ↓ oxidative stress ↓ Alzheimer’s disease (AD) symptoms	[30]
CNPs	NA (purchased)	Spherical	ND	ND	C57BL/6J male mice exposed to PM2.5 for 6 h/day, 5 times a week for 23 weeks as *in vivo* model for Alzheimer’s disease (AD)	0.25 mg/kg or 0.5 mg/kg, 1 time a week for 10 weeks	Intravenous tail injection	↓ IL-6, TNF-α, IL-1β ↓ NF-κBia and IκBκB ↓ GFAP ↓ ROS (O_2_, H_2_O_2_) ↑ SOD activity	↓ inflammation ↓ glial cell activation ↓ oxidative stress ↑ antioxidant response	[32]
					Mice-derived primary astrocytes treated with PM2.5 for 24 h	5–200 ng/mL for 24 h	NA	↓ GFAP ↓ IL-6, TNFα, IL-1β ↓ nuclear NF-κB ↑ cytoplasmatic NF-κB ↓ COX-2 ↓ ROS (O_2_^−^, H_2_O_2_), iNOS ↑ SOD1, SOD2, NQO1	↓ glial cell activation ↓ inflammation ↓ oxidative stress ↑ antioxidant response	
CNPs	NA (purchased)	ND	ND	3.3	SOD1^G93A^ transgenic mouse as *in vivo* model for amyotrophic lateral sclerosis (ALS)	20 mg/kg, 2 times a week	Intravenous tail injection		↑ muscle strength ↑ survival (+11 days ca.) ↓ clinical score ↓ body weight loss	[33]
CNPs	NA (purchased)	Spherical	3–5	ND	Wistar rats exposed to unpredictable chronic mild stress (UCMS) as *in vivo* model for stress-induced depression	10 nM	Intrahippocampal and intracerebroventricular injection	↓ IL-6 ↓ MDA ↑ GAP-43+ neurons ↑ CA3 neurons number	↓ depressive-like behavior ↓ inflammation ↓ oxidative stress ↑ neurogenesis ↑ neurite outgrow	[35]
CNPs	Reverse micelle	Spherical	3.5 ± 0.5	ND	VC57BL/6J male mice exposed to open head injury using a stereotaxic impactor as *in vivo* model for traumatic brain injury (TBI)	11.6 mM	Retro-orbital injection	↓ FJB^+^ and TUNEL^+^ cells ↑ SOD1 and SOD2 mRNA ↓ COX-2	↓ neuronal cell death ↓ oxidative stress ↓ inflammation ↑ cognitive functions ↓ cerebral edema	[36]
	Hydrothermal	Rods	9.4 ± 2.1 Length: 130.1 ± 42.1	ND						

CNPs: cerium oxide nanoparticles; ROS: reactive oxygen species; A β: amyloid-beta; β-TubIII: beta-tubulin III; GAP-43: growth-associated protein 43; NF-H200: neurofilament 200; SOD: superoxide dismutase; BDNF: brain-derived neurotrophic factor; TrkB: BNDF receptor; GPX: glutathione peroxidase; p-ERK: phospho- extracellular signal-regulated kinase; GFAP: glial fibrillary acidic protein; Iba-1: ionized calcium-binding adapter molecule 1; 4-HNE: 4-hydroxynonenal; IL: interleukin; TNF-α: tumor necrosis factor-alpha; p-NF-κB: phospho-nuclear factor kappa-light-chain-enhancer of activated B cells; COX: cyclooxygenase; iNOS: inducible nitric oxide synthase; NQO1: NAD(P)H quinone dehydrogenase 1; NADPH: nicotinamide adenine dinucleotide phosphate; MDA: mondialdehyde; GAP-43: growth-associated Protein 43; CA3: hippocampal region; FJB: fluoro-jade B staining; FAK: focal adhesion kinase; STAT3: transducer and activator of transcription 3; CD36: cluster of differentiation 36; Rab7: Ras-related protein 7; LAMP1: lysosomal-associated membrane protein 1; NA: not applicable; ND: not determined. Colored areas highlight *in vivo* experiments.

**Table 2 nanomaterials-13-02803-t002:** CNPs in inflammatory autoimmune diseases: literature summary.

Nanomaterial	Synthesis Method	Morphology	Size (nm)		Model	Dosage	Route of Administration	Markers	Effects	Ref.
			Powder	Hydrodynamic Radius						
Albumin-conjugated CNPs (A-CNP)	Biomineralization	Spherical	30 ± 8.9	ND	DBA/1J mice exposed to collagen-induced arthritis (CIA) mouse as *in vivo* model for rheumatoid arthritis	1 mg/kg, 2 times a week	Intra-articular injection	↓ iNOS ↑ Arg-1	M2 polarization ↓ clinical score	[39]
					THP-1 human monocytes and RAW 264.7 murine macrophages	0.5 μg/mL for 24 h	NA	↓ iNOS, IL-1β ↑ Arg-1 ↓ HIF-1α	M2 polarization ↑ antioxidant response ↓ hypoxia	
Citric-acid-coated CNPs (CA-CNPs)	Alkaline-based precipitation	Spherical	2.8 ± 0.4	3.4 ± 1.1	HepG2 hepatocyte cells, RAW264.7 murine macrophages, Renca epithelial kidney cells and SVEC4-10EHR1 endothelial cells exposed to LPS or H_2_O_2_	0.1–1 mg/mL for 24 h	NA	↑ SOD, CAT and HORAC activity ↓ ROS (OH^•^) ↓ TNF-α, IL-1β	↑ cell viability ↓ oxidative stress ↓ inflammation	[40]
					C57BL/6J mice treated with complete Freund’s adjuvant (CFA) as *in vivo* model for peripheral Inflammation	100 mg/kg	Intravenous tail injection	↓ TNF-α, IL-1β ↑ IL-10	↓ paw inflammation ↓ edema formation ↓ immune cell infiltration	
CNPs + Hyaluronic acid	hydrothermal	Cubic	10–60	ND	Chondrocytes treated with H_2_O_2_ for 30 min	0.02 μg/mL	NA	↑ ACAN, COL1A1, COL2A1	↓ cell apoptosis ↓ oxidative stress ↓ glycosaminoglycan synthesis	[42]
CNPs	NA	Spherical	5	10	Sprague-Dawley-rat-derived chondrocytes treated with IL-1β	160 μg/mL	NA	↓ ROS (O_2_^−^) ↓ NO ↑ Nrf2, HO-1, SOD, CAT, GPX ↑ ACAN, COL1A1, COL2A1 ↓ MMP13, ADAMTS4 ↓iNOS, COX-2, IL-6	↓ oxidative stress ↑ antioxidant response ↓ cell apoptosis ↓ ECM degradation ↓ inflammation	[43]
					Sprague-Dawley-rat-derived condylar cartilage explants treated with IL-1β			↓ ROS (O_2_^−^) ↓ NO	↓ apoptosis ↓oxidative stress	
CNPs + Lenalidomide	Precipitation	Spherical	3–5	34 ± 6.8	C57BL/6 mice treated with MOG 35–55 peptide and pertussis toxin (experimental autoimmune encephalomyelitis) as *in vivo* model for multiple sclerosis (MS)	1 mg/kg, every fourth day	Intravenous injection	↑ MBP ↓ TNF-α ↓ IL-17, INF-γ, TNF-α (mRNA) ↓ GFAP, Iba-1 ↓ CD86^+^ dendritic cells	↓ clinical score ↑ body weight ↓ ventricular volume ↓ myelin loss ↓ inflammation ↓ glial cell activation ↓ peripheral immune reaction	[45]

CNPs: cerium oxide nanoparticles; iNOS: inducible nitric oxide synthase; M1/2: macrophage 1/2; Arg-1: arginase-1; HIF-1α: hypoxia inducible factor-1 alpha; SOD: superoxide dismutase; CAT: catalase; HORAC: hydroxyl radical antioxidant capacity (HORAC) assay; ROS: reactive oxygen species; IL: interleukin; MBP: myelin basic protein; INF-γ: interferon gamma; TNF-α: tumor necrosis factor-alpha; GFAP: glial fibrillary acidic protein; Iba-1: ionized calcium-binding adapter molecule 1; ACAN: aggrecan; COL: collagen; NO: nitrite; HO-1: heme oxygenase 1; GPX: glutathione peroxidase; MMP: matrix metallopeptidase; ADAMTS4: metallopeptidase with thrombospondin type 1 motif 4; COX: cyclooxygenase; ECM: extracellular matrix; Nrf2: nuclear factor erythroid 2-related factor 2; ND: not determined; NA: not applicable. Colored areas highlight *in vivo* experiments.

**Table 3 nanomaterials-13-02803-t003:** CNPs in liver inflammation: literature summary.

Nanomaterial	Synthesis Method	Morphology	Size (nm)		Model	Dosage	Route of Administration	Markers	Effects	Ref.
			Powder	Hydrodynamic Radius						
CNPs	ND	ND	ND	ND	Rats treated with monosodium glutamate (MSG) as *in vivo* model for non-alcoholic fatty liver disease (NAFLD)	1 mg/kg 1 time for month	Oral gavage	↓ total lipids, triglycerides ↓ IL-1β, IL-12Bp40 ↑ TGF-β, IL-4 ↓ IL-10	↓ liver damage ↓ NAFLD activity score ↓ inflammation ↓ obesity	[48]
CNPs	NA (purchased)	Spherical	10–30	ND	Male Wistar rats treated with carbon tetrachloride (CCl4) as *in vivo* model for non-alcoholic fatty liver disease (NAFLD)	0.1 mg/kg 2 times a week for 2/4 weeks	Intravenous injection	↑ TAC ↑ GSH ↓ TNF-α Normalization of ALP, ALT, and AST levels ↓ MDA	↑ antioxidant capacity ↓ inflammation ↓ CCl_4_-induced liver injury	[49]
CNPs	NA (purchased)	Spherical and cuboidal	10–25	ND	Female Wistar rats subjected to ovariectomy operation as *in vivo* model for postmenopausal obesity	0.1 mg/kg 2 times a week for 2 weeks	Intraperitoneal injection	↓ LXR, AST, ALT ↓ FFA, TG, TC, LDL-C ↑ HDL-C ↓ MDA, TAC ↓TNF-α, TGF-1β	↓ obesity ↓ steatosis ↓ lipogenesis ↓ oxidative stress ↓ inflammation	[50]
CNPs	Chemical precipitation	Spherical	4–20	ND	Carbon tetrachloride (CCl4)-treated rats as *in vivo* model for liver fibrosis	0.1 mg/kg 2 times a week for 2 weeks	Intravenous tail injection	↓ AST, ALT, α-SMA ↓ CD68^+^ cells ↓ TUNEL^+^ cells ↓ activated- caspase 3 ↓ IL-1β, TNF-α, COX-2, iNOS ↓ Ncf1, Ncf2, Epx	↓ steatosis ↓ fibrogenesis ↓ apoptotic cell death ↓ inflammation ↓ oxidative stress	[52]
					HepG2 hepatocyte cells treated with H_2_O_2_	100 μg/mL	NA	↓ ROS	↓ oxidative stress	
CNPs	NA (purchased)	Cubical	120 ± 7.5	ND	Male C57BL/6J mice subjected to bile duct ligation (BDL) as *in vivo* model for liver fibrosis	0.5 mg/kg or 2 mg/kg for 12 days	Intraperitoneal injection	↓ SGOT, SGPT, ALP, bilirubin ↓ MDA ↓ nitrite level ↑ SOD, CAT, GSH ↓ IL-1β, IL-6, IL-17, TNF-α, TGF-β ↓ p65-NF-κB, COX-2, iNOS ↓ Snail, TIMP-1, α-SMA, LOXL-2, N-Cad, fibronectin	↓ liver fibrosis ↓ oxidative stress ↓ nitrative stress ↑ antioxidant response ↓ liver fibrosis ↓ inflammation	[53]
CNPs	NA (purchased)	Spherical	10–30	70	Male Sprague-Dawley rats as *in vivo* model for hepatic ischemia reperfusion (IR)	0.5 mg/kg 20 min before IR	IV tail injection	↓ ALT, LDH ↓ KC/GRO, MDC, MIP-2, myoglobin, leptin, insulin, PAI-1, vWF (inflammatory mediators) ↑ growth hormone	↓ liver damage ↓ hepatocyte necrosis ↓ inflammation	[54]
CNPs	ND	ND	ND	ND	Male mice as *in vivo* model for hepatic ischemia reperfusion (IR)	300 μg/kg 24 h before IR	Intraperitoneal injection or oral gavage	↓ LDH, ALT, AST ↓ MDA ↑ GSH ↓ lipid peroxidation ↑ SOD, CAT, GPx ↓ p65-NF-κB, MPO activity ↓ TNF-α, IL-1α, IL-1β, IL-2, IL-6, IL-8, IL-12, IL-17, ICAM-1 ↑ IL-10 ↓ MMP-2, MMP-9, TIMP-1	↓ liver edema ↓ liver injury ↓ oxidative stress ↑ antioxidant response ↓ inflammation ↓ MPP activation	[56]

CNPs: cerium oxide nanoparticles; IL: interleukin; TGF-β: transforming growth factor beta; LXR: liver X receptor; AST: aspartate aminotransferase; ALT: alanine transaminase; FFA: free fatty acid; TG: triglycerides; TC: total cholesterol; LDL-C: low-density lipoprotein cholesterol; HDL-C: high-density lipoprotein cholesterol; MDA: mondialdehyde; SOD: superoxide dismutase; CAT: catalase; TAC: total antioxidant capacity; TNFγ: tumor necrosis factor gamma; α-SMA: alpha smooth muscle actin; SGOT: serum glutamic oxaloacetic transaminase; SGPT: serum glutamic pyruvic transaminase; ALP: alkaline phosphatase; GSH: glutathione; GPX: glutathione peroxidase; NF-κB: nuclear factor kappa-light-chain-enhancer of activated B cells; COX: cyclooxygenase; iNOS: inducible nitric oxide synthase; TIMP-1: tissue inhibitor of metalloproteinase 1; LOXL-2: lysyl oxidase like 2; N-Cad: N-cadherin; LDH: lactate dehydrogenase; KC/GRO: keratinocyte chemoattractant/human growth-regulated oncogene; MDC: macrophage-derived chemokine; MIP-2: macrophage inflammatory protein-2; PAI-1: plasminogen activator inhibitor-1; vWF: von Willebrand factor; GSH: glutathione; MPO: myeloperoxidase; ICAM-1: intercellular adhesion molecule-1; MMP: matrix metalloproteinase; NA: not applicable; ND: not determined. Colored areas highlight the *in vivo* experiments.

**Table 4 nanomaterials-13-02803-t004:** CNPs in gastrointestinal inflammatory disorders: literature summary.

Nanomaterial	Synthesis Method	Morphology	Size (nm)		Model	Dosage	Route of Administration	Markers	Effects	Ref.
			Powder	Hydrodynamic Radius						
CNPs	Precipitation	ND	160	ND	Female Sprague-Dawley rats treated with 90% ethanol as *in vivo* model for gastric ulcers	1 mg/kg	Oral gavage	↑ SOD and CAT activity	↓ ulcerative lesions ↑ antioxidant response	[59]
Citrate-coated CNPs	Sol-gel	ND	3–7	4.9	Male albino nonlinear rats exposed to ulcerogenic factor as *in vivo* model for gastric ulcers	1 mg/kg 24 h before exposure to ulcerogenic factor	Oral gavage	↓ lipid peroxidation ↓ ROS (H_2_O_2_) ↑ SOD activity ↓ CAT activity ↓ IL-1β, IL-12Bp40, INF-γ ↑ IL-4, IL-10, TGF-β	↓ ulcerative lesions ↑ protective protein of gastric mucosa (GM) ↓ inflammation ↓ oxidative stress ↑ antioxidant response	[60]
Sulfasalazine-linked NH_2_-CNPs (SSZ@NH_2_-CNPs)	UiO-66 (Ce) synthetic	Semi-spherical	64.9 ± 15.6	ND	C57BL/6 male mice treated with 1.5% (*w*/*v*) dextran sodium sulphate (DSS) for 7 day as *in vivo* model for ulcerative colitis	258 mg/kg/day for 7 days	Oral gavage	↓ MDA ↓ collagen deposition ↓ lipid peroxidation ↑ SOD and CAT activity ↑ total thiols level	↓ disease activity index ↑ body weight ↓ colon shortening and spleen weight ↓ inflammation ↓ oxidative stress ↑ antioxidant response ↓ fibrosis	[61]

CNPs: cerium oxide nanoparticles; SOD: superoxide dismutase; CAT: catalase; ROS: reactive oxygen species; IL: interleukin; INF-γ: interferon gamma; TGF-β: transforming growth factor beta; MDA: mondialdehyde; ND: not determined; NA: not applicable. Colored areas highlight *in vivo* experiments.

**Table 5 nanomaterials-13-02803-t005:** CNPs in ocular inflammation: literature summary.

Nanomaterial	Synthesis Method	Morphology	Size (nm)		Model	Dosage	Route of Administration	Markers	Effects	Ref.
			Powder	Hydrodynamic Radius						
CNPs	ND	ND	ND	ND	Rat-derived primary retinal neurons exposed to 1 mM H_2_O_2_ for 30 min	1–20 nM	NA	↓ ROS	↓ oxidative stress	[71]
					Sprague-Dawley albino rats exposed to 2700 lux of light for 6 h as *in vivo* model for light-induced photoreceptor degeneration	1–20 nM pre- and post-light damage induction	Intravitreal injection	↑ Thickness of ONL ↓ TUNEL^+^ cells	Protection of retina photoreceptor ↓ retinal degeneration ↓ photoreceptor cell apoptosis ↑ retinal function	
CNPs	ND	ND	3–5	ND	Vldlr^−/−^ mutant mice exposed to 80 lux of light as *in vivo* model for age-related macular degeneration (AMD)	1 mL of 1 mM (172 ng) of CNPs	Intravitreal injection	↓ ROS ↓ VEGF-A ↓ p-ERK1/2, p-JNK1/2, p-p38, p-Akt	↓ vascular lesion ↓ angiogenesis ↓ inflammation	[72]
CNPs	NA (purchased)	ND	10–100	ND	Human corneal epithelial cells (HCECs) and RAW264.7 murine macrophages treated with H_2_O_2_	2–80 μg/mL	NA	↓ ROS, NO ↓ TNF-α, IL-6	↓ oxidative stress ↓ inflammation	[73]
					Sprague-Dawley albino rats and adult Japanese white rabbits exposed to ocular alkali burns as *in vivo* models for corneal neovascularization	20 μL of 200 mg/mL of CNPs	Dropping into the lower conjunctival sac	↓ TNF-α	↓ corneal opacification ↓ corneal neovascularization ↓ inflammation	
CNPs	Hydrolysis and condensation of Ce(Cit)_2_^−3+^ at high pH	Spherical	3	4.3	ARPE19 retinal cells and HUVEC human vascular endothelial cells	5–500 nM	NA	↓ ROS ↑ SOD expression ↓ VEGFA ↓ microvessels	↑ antioxidant response ↓ cell migration and neovascularization	[74]
					DKOrd8 mouse model as *in vivo* model of dry AMD-like pathology and laser-induced choroidal neovascularisation (LI-CNV) mouse model	2 mg/mL 2-months daily treatment	Intravitreal and topical administration	↓ number of lesions in photoreceptors layers ↑ Nrf2 ↓ microglia cell number ↓ IL-18	↓ oxidative stress ↑ retinal function ↑ DNA repair ↑ senescence	

CNPs: cerium oxide nanoparticles; ROS: reactive oxygen species; ROI: reactive oxygen intermediates; ONL: outer nuclear layer; Vldlr: very low-density lipoprotein receptor; Akt: anti-apoptotic serine/threonine kinase; GF: growth factor; VEGFA: vascular endothelial growth factor A; IL: interleukin; ERK: extracellular signal-regulated kinase; NO: nitric oxide; TNF-α: tumor necrosis factor alpha; NA: not applicable; ND: not determined;. Colored areas highlight *in vivo* experiments.

**Table 6 nanomaterials-13-02803-t006:** CNPs in pathogen-induced inflammation: literature summary.

Nanomaterial	Synthesis Method	Morphology	Size (nm)		Model	Dosage	Route of Administration	Markers	Effects	Ref.
			Powder	Hydrodynamic Radius						
CNPs immobilized on the surface of silica NPs (S-CNPs)	Precipitation	ND	220 ± 5	ND	Wistar male rats treated with 1 mg/kg LPS as *in vivo* model for experimental pneumonia	0.6 mg/kg at 0, 1, 3 and 24 h after LPS injection	Orogastric catheter	↓ ROS (O_2_^•- •^OH, H_2_O_2_) ↓ TNF-α, IL-6 ↓ CXCL2	↓ lung injuries ↓ oxidative stress ↓ inflammation ↑ oxygen consumption	[75]
CNPs	NA (purchased)	Spherical	ND	140 ± 53	Sprague-Dawley male rats subjected to intraperitoneal injection of cecal material (400 mg/kg) as *in vivo* model for polymicrobial sepsis	3.5 mg/kg	Intravenous injection	↓ IL-6 ↓ blood urea nitrogen	↑ animal survival ↓ inflammation ↓ liver and renal dysfunction	[78]
					RAW264.7 murine macrophages treated with LPS (2 μg/mL) for 24 h	0.72–8.6 μg/mL for 24 h	NA	↓ ROS ↓ TNF-α, IL-1β, IL-6 ↓ iNOS, COX-2 ↓ nuclear NF-κB ↑ cytoplasmic NF-κB	↑ cell survival ↓ oxidative damage ↓ inflammation	
					Gram-negative *Escherichia coli* and Gram-positive *Staphylococcus aureus*	1–100 mg/mL	NA		↓ bacterial growth (only for *E. coli*)	
CNPs	NA (purchased)	Spherical	200–400	53.36 ± 7.04	Sprague-Dawley rats subjected to intraperitoneal injection of LPS as *in vivo* model of hepatic dysfunction	0.5 mg/kg	Intravenous tail injection	↓ TNF-α, IL-1β, IL-1α ↓ bilirubin, ALT, GST-Mu, GST-α ↓ MyD88, p-p38-MAPK, p-p44/42-MAPK, p-ERK1/2 ↓ iNOS, HMBG-1 ↓ cleaved caspase 3 ↓ Bax/Bcl-2 ratio ↓ ROS	↑ animal survival ↑ blood pressure ↓ immune cell infiltration ↓ inflammation ↓ hepatic damage ↓ apoptosis ↓ oxidative stress	[83]
					RAW264.7 murine macrophages treated with LPS	0.1–1000 ng/mL	NA	↓ ROS ↓ TNF-α, IL-1β, IL-6 ↓ iNOS, HMBG-1 ↓ COX-2 ↓ nuclear NF-κB	↓ cell apoptosis ↓ oxidative stress ↓ inflammation	
CNPs	NA (purchased)	ND	ND	ND	Sprague-Dawley male rats subjected to intravenous tail injection of LPS (40 mg/kg) as *in vivo* model for severe sepsis	0.5 mg/kg	Intravenous tail injection	↓ HMGB1	↓ splenic damage ↓ inflammation ↓ bacterial load in blood and peritoneal fluid	[84]
CNPs	NA (purchased)	Cubic	10–30	90	Male Sprague-Dawley rats subjected to intraperitoneal injection of cecal material (600 mg/kg) as *in vivo* model for polymicrobial sepsis	0.5 mg/kg	intravenous tail injection	↓ ROS (superoxide levels) ↓ iNOS, nitrotyrosine ↓ TNF-α, INF-γ, IL-6, GST-α, GST-μ ↓ p-ERK1/2, p-Stat-3 ↓ P-selectin, VCAM-1	↑ animal survival ↓ animal hypothermia ↓ oxidative stress ↓ hepatic damage ↓ nitrosative stress ↓ inflammation ↓ immune cell infiltration	[85]
					RAW264.7 murine macrophages	1–100 ug/mL for 48 h	NA		Non-toxic effect	
CNPs	NA (purchased)	Spherical	15–20	ND	Male Sprague-Dawley rats subjected to intraperitoneal injection of cecal material (600 mg/kg) as *in vivo* model for polymicrobial sepsis	0.5 mg/kg	Intravenous tail injection	↓ p-Stat-3 ↓ iNOS ↑ p-Akt, p-FOXO-1, p-4EBP1 ↓ caspase-8 cleavage	↑ diaphragmatic function ↓ cell infiltration ↓ inflammation ↓ oxidative stress ↓ protein degradation	[87]
6-aminohexanoic acid (6-AHA)-conjugated CNPs (6-AHA-CNPs)	Sol–gel	Spherical and square	25	35	Male C57BL/6 mice subjected to cecal ligation puncture peritonitis as *in vivo* model for systemic inflammatory response syndrome (SIRS)	0.5 mg/kg	Intravenous tail injection	↓ MPO, CD68^+^ cells	↓ lung and liver injuries ↓ hepatocellular necrosis ↓ immune cell infiltration ↓ inflammation ↑ animal survival	[88]
					RAW 264.7 murine macrophages and U937 human leukemic monocytes treated with H_2_O_2_ or LPS	0.1 mM	NA	↓ ROS (O_2_^−^, H_2_O_2_) ↓ IL-1β, LDH ↓ iNOS	↓ oxidative stress ↓ inflammation	

CNPs: cerium oxide nanoparticles; NPs: nanoparticles; LPS: lipopolysaccharide; ROS: reactive oxygen species; TNF-α: tumor necrosis factor alpha; IL: interleukin; CXCL2: chemokine (C-X-C motif) ligand 2; iNOS: nitric oxide synthases; NF-κB: nuclear factor kappa-light-chain-enhancer of activated B cells; ROS: reactive oxygen species; HMGB1: high mobility group box 1 protein; GST: glutathione S-transferase; IFN-γ: interferon gamma; p-ERK: phosphorylated-extracellular signal-regulated kinase; p-Stat-3: phosphorylated-signal transducer and activator of transcription 3; VCAM-1: vascular cell adhesion molecule 1; p-AKT: phosphorylated-protein kinase; p-FOXO1: phosphorylated-forkhead box protein O1; p-4EBP1: phosphorylated eukaryotic translation initiation factor 4E-binding protein 1; MPO: myeloperoxidase; LDH: lactate dehydrogenase; ALT: alanine aminotransferase; GST: glutathione S-transferase;MyD88: myeloid differentiation primary response 88; MAPKs: mitogen-activated protein kinases Bax: Bcl-2-associated X-protein; Bcl-2: B-cell lymphoma 2; NA: not applicable; ND: not determined;. Colored areas highlight *in vivo* experiments.

**Table 7 nanomaterials-13-02803-t007:** CNPs in regenerative medicine: literature summary.

Nanomaterial	Synthesis Method	Morphology	Size (nm)		Model	Dosage	Route of Administration	Markers	Effects	Ref.
			Powder	Hydrodynamic Radius						
CNPs	Hydrothermal	Cubic	19.5	33.7	Sprague-Dawley-rat-derived primary cortical neurons treated with H_2_O_2_	1–4000 μg/mL for 24 h	NA	↓ iNOS	↑ neuronal cell survival ↓ oxidative stress	[96]
					Sprague-Dawley rats exposed to contusion injury at T9 level as *in vivo* model for spinal cord injury (SCI)	0.5–4 mg/mL	Lesion cavity injection	↓ ED1+ cells ↓ iNOS ↓ Nr-f2, COX-2 ↓ TNF-α, IL-1β, IL-6 ↑ IL-10 ↓ p53, caspase 3	↓ lesion cavity ↓ inflammation ↓ oxidative stress ↑ locomotor functions ↓ apoptosis ↑ axonal regeneration	
BSA-incubated-CNP-based hydrogel (CNPs-Gel)	BSA-incubation method	Spherical	ND	4.97 ± 1.39	Sprague-Dawley rats exposed to injury at T10 level as *in vivo* model for spinal cord injury (SCI)	3–6 mM	Implantation of NSC-loaded CNP gel into the lesion	↓ ROS, 4-HNE ↓ CD68^+^ cells ↓ iNOS ↑ Arg-1 ↑ p-FAK, p-PI3K, p-AKT1 ↓ GFAP ↑ MAP-2 ↑ NF^+^ and MBP^+^ cells	↑ motor function ↓ cavity area ↓ oxidative stress ↓ inflammation M2 polarization ↓ glial cell activation ↑ neuron and axonal regeneration	[98]
					BV2 microglial cell and neuronal stem cells (NSCs) treated with H_2_O_2_	ND	NA	↓ ROS ↓ TNF-α, iNOS, IL-6 ↑ IL-10, TGF-β, Arg-1 ↑ p-FAK, p-PI3K, p-AKT1	↑ cell survival ↓ oxidative stress ↓ inflammation M2 polarization	
CNPs	Wet chemical	Spherical	5	35	RAW 264.7 murine macrophages and human bone mesenchymal stem cells (BMSC) treated with LPS	1, 10, or 20 μg/mL	NA	↓ iNOS ↓ IL-6, IL-1β, TNF-α ↑ IL-10, TGF-β	↓ inflammation ↑ osteogenic differentiation	[99]
CNPs incorporated hydroxyapatite coating	Plasma spraying	ND	ND	ND	Rat bone mesenchymal stem cells and RAW264.7 macrophages	10–30 wt%	NA	↑ ALP, OCN, Runx-2 ↑ BMP2, BMPR1, BMPR2, Smad1, Smad5, and Smad8 ↓ TNF-α, IL-6 ↓ CCR7 and CD11c ↑ CD163, CD206 ↑ IL-1rα, IL-10, TGF-β	↑ cell proliferation M2 polarization ↑ osteogenic differentiation ↓ inflammation	[100]
Immobilized CNPs on titanium-based biomaterial	Deposition by magnetron sputtering and vacuum annealing	ND	ND	ND	Rat bone marrow mesenchymal stem cells (BMSCs), RAW264.7 murine macrophages	Elemental concentration: 3.57–7.58%	NA	↑ ALP, Col-I, OCN, OPN, Runx-2 ↑ IL-10 ↓ TNF-α	↑ cell proliferation ↑ osteogenic differentiation M2 polarization	[101]
					Sprague-Dawley rats		Femoral bone implantation		Femoral bone implantation	
CNP-modified titanium disks (CNPs@TiO_2_)	Hydrothermal	Nanorods	9.6 ± 1.2 Length: 50–600	ND	Gram-positive bacteria *S. sanguinis* and Gram-negative bacteria *F. nucleatum*	0.1 M	NA		↓ bacterial growth	[103]
		Cubic	57.2 ± 17.5	ND	RAW 264.7 murine macrophages treated with LPS		NA	↓ ROS ↓ TNF-α, IL-1β, IL-6 ↓ nuclear NF-κB (NF-κB/p65)	↓ oxidative stress ↓ inflammation	
		Octahedral	29.1 ± 11.7	ND	Wistar rats		Subcutaneous implantation	↓ TNF-α, IL-1β, IL-6	↓ immune cell infiltration ↓ inflammation	
CNPs	NA	spherical	5	10	RAW 264.7 cells	2–50 μg/mL	NA	↓ MAPK NFkB pathway ↓ IL-1β ↓ TNF-α ↓ iNOS ↓ Nrf2, HO-1 ↓ ROS	↓ oxidative stress ↓ inflammation	[104]
					Male Sprague-Dawley rats treated with LPS as *in vivo* model of gingival inflammation	10 μL; 2 mg/mL	Gingival injection	↓ ROS	↓ destruction of periodontal tissues	

CNPs: cerium oxide nanoparticles; iNOS: nitric oxide synthases; Nr-f2: nuclear factor erythroid 2-related factor 2; COX: cyclooxygenase; TNF-α: tumor necrosis factor alpha; IL: interleukin; BSA: bovine serum albumin; M: macrophages; ROS: reactive oxygen species; 4-HNE: 4-hydroxynonenal; Arg-1: arginase 1; p-FAK: phosphorylated-focal adhesion kinase; p-PI3K: phosphorylated-phosphatidylinositol 3-kinase; p-AKT1: phosphorylated-protein kinase B; GFAP: glial fibrillary acidic protein; MAP2: microtubule-associated protein 2; NF: neurofilament protein; MBP: myelin basic protein; LPS: lipopolysaccharide; ALP: alkaline phosphate; OCN: osteocalcin; Runx-2: runt-related transcription factor 2; BMP2: bone morphogenetic protein 2; BMPR: bone morphogenetic protein receptor; Smad: suppressor of mothers against decapentaplegic; CCR7: C-C chemokine receptor type 7; Col-I: collagen type I; OPN: osteopontin; NF-κB: nuclear factor kappa-light-chain-enhancer of activated B cells; HO-1: hemeoxygenase-1; ND: not determined; NA: not applicable. Colored areas highlight *in vivo* experiments.

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
