# Peer review of "The Impressive Anti-Inflammatory Activity of Cerium Oxide Nanoparticles: More than Redox?"

_nanomaterials, 2023, doi:10.3390/nano13202803_

Round 1
Reviewer 1 Report
The paper presents a review of the current literature devoted to the anti-inflammatory activity of cerium oxide nanoparticles. The subject of the paper fits the scope of Nanomaterials journal.
I have the following comments:
1. Currently, ceria-based materials are among the most well-studied nanobiomaterials. Unfortunately, the list of references used in this review is quite short: it includes 92 papers only, with 8 papers written by the authors of the review. Moreover, 30 papers of 92 listed in References section were published more than 10 ago, while the number of papers revealing the very special properties of ceria increased exponentially in the last few years. Surely, this doesn't allow the authors to show a whole up-to-date picture.
2. Bioactivity of ceria, including (but not limited by) its anti-inflammatory activity, was already discussed in many similar reviews. What is the disctinct reason for preparing another review on this topic?
3. Recent papers discussing the reasons underlying ceria bioactivity are largely ignored. For instance, only few types of enzyme-like activity of ceria are discussed in the review. The existing data on the up- and down-regulation of genes expression by ceria nanmoparticles are almost ignored. The reasons underlying the anti-inflammatory activity of ceria are not unveiled.
4. The major part of the review presents a mix of brief excerpts from the existing papers, which are not discussed at all. No logical connections can be traced between these excerpts. Thus the review doesn't provide an integral and concise view on the problem of anti-inflammatory activity of ceria.
5. In numerous tables, the information on the particle size and morhpology is provided which is not discussed at all. Is this an attempt to elucidate size effect? Please explain. In the "Synthesis method" column, the following remark can be frequently found: "Purchased from Sigma-Aldrich". Surely, this is not a synthesis method, and this tells nothing about the method for producing these nanoparticles.
6. It is well established that nano-ceria act as bot anti- and prooxidant. Unfortunately, the latter function is not discussed by the authors while it is of high relevance to the scope of the review. For instance, when discussing the action of nano-ceria on gastro-intestinal inflammatory disordes, one should take into account the high acidity of the gastric juice. Under these conditions, ceria becomes a strong pro-oxidant. Moreover, in acidic conditions nano-ceria readily dissolves to produce Ce3+ species.
7. In Lines 73-74, the authors mentioned so called "complete energy-free redox cycle". What is this? Please provide the corresponding values of Gibbs free energy changes to confirm that this cycle is actually energy-free.
To summarize, this review is not suitable for publication in Nanomaterials in its present form.
Reviewer 2 Report
The review examines numerous literature data obtained using in vitro and in vivo experiments, which indicate the anti-inflammatory properties of cerium oxide nanoparticles obtained by various methods. The review examines various clinical and socially significant pathologies, such as reimplantitis that occurs during the healing of dental implants, inflammation due to traumatic injuries obtained by tissue engineering methods, pathogen-induced inflammation, eye pathologies, gastrointestinal inflammatory diseases, liver inflammation, inflammatory autoimmune diseases , which these nanoparticles are capable of influencing, producing a positive effect. The review contains many tables that briefly and clearly summarize the experimental data presented in the articles for each section. This form of presenting results is convenient for perceiving information.
As a side note, the authors could have included more articles published in the last two years in the reference list.

Author Response
We thank reviewer 2 for her/his favorable comment. As far the side note, we have expanded the number of articles presented and commented, including articles published in the last 3 years that we have missed in the previous version.
Reviewer 3 Report
The theme of the review manuscript is novel, the summary is comprehensive and detailed, however, the single words and tables looks boring to readers I’d recommend add some figures if possible, here are some other suggestions:
Major:
1. I think the following significant paper should be cited, as far as I know, this is the first paper to state the anti-inflammatory property of CNPs.
Anti-inflammatory Properties of Cerium Oxide Nanoparticles, small 2009, 5, No. 24, 2848–2856, https://doi.org/10.1002/smll.200901048
2. Here, the authors state cerium oxide nanoparticles can exhausted ROS, however, in some other works (like this: Cerium Oxide Nanoparticles Induced Toxicity in Human Lung Cells: Role of ROS Mediated DNA Damage and Apoptosis, Volume 2014 | Article ID 891934 | https://doi.org/10.1155/2014/891934 ), the researcher found it will produce ROS and kill the cells, could the authors gave some analysis and statements?
3. The structure should be clearer, I’d suggest set as this:
1. Introduction
2. Anti-inflammatory application of cerium oxide nanoparticles
2.1. Neurodegenerative diseases
2.2. Inflammatory autoimmune diseases
2.3. Liver inflammation
2.4. Gastro-intestinal inflammatory disorders
2.5. Ocular inflammation
2.6. Pathogen-induced inflammation
2.7. Inflammation in tissue engineering-treated traumatic injuries: premises and promises of CNPs
3. Conclusions: anti-inflammatory = antioxidant?
4. Future Directions
4. I’d suggest supplementing the mechanism of nanozymes activity paragraph of CNPs and show the illustration scheme.
Minor:
1. Line 36, singlet oxygen should be included.
2. Line 46, what converts 2 H2O2 to 2 H2O + O2?
3. Line 47, where the GSH system exists should be mentioned.
4. Line 61, I don’t agree with the statement in the brackets, that’s not a definition of nanoparticles.
5. Line 345, in vivo should be italic.
6. Page 4 should be deleted.
